# Influence of Perineural (Pn), Lymphangio (L) and Vascular (V) Invasion on Survival after Resection of Perihilar Cholangiocarcinoma

**DOI:** 10.3390/cancers16203463

**Published:** 2024-10-12

**Authors:** Rabea Margies, Lisa-Katharina Gröger, Beate K. Straub, Fabian Bartsch, Hauke Lang

**Affiliations:** 1Department of General, Visceral and Transplant Surgery, University Medical Center, Johannes Gutenberg-University Mainz, Langenbeckstraße 1, 55131 Mainz, Germany; rabea.margies@unimedizin-mainz.de (R.M.);; 2Department of Pathology, University Medical Center, Johannes Gutenberg-University Mainz, Langenbeckstraße 1, 55131 Mainz, Germany

**Keywords:** perihilar cholangiocarcinoma, overall survival, perineural invasion, lymphangio invasion, vascular invasion, liver surgery

## Abstract

**Simple Summary:**

Perihilar cholangiocarcinoma is a rare biliary cancer with a poor prognosis, even after surgery. This study aimed to assess the impact of perineural (Pn), lymphangio (L), and vascular (V) invasion on overall survival following surgical resection. Data from 214 patients who underwent surgery between 2013 and 2023 were analyzed. Of these, curative intended resection was achieved in 168 patients (78.5%). Perineural, lymphangio and vascular invasion were present in 79.2%, 17.3% and 14.3% of patients, respectively. L1 and V1 were significantly associated with each other. Vascular invasion was associated with a significantly worse overall survival, while perineural invasion showed a trend toward worse overall survival, although this was not statistically significant. In Bismuth type IV tumors, both L1 and V1 were significantly linked to poorer overall survival. Furthermore, the combination of Pn1 with the presence of either L1 or V1 (or both) was associated with worse overall survival, underscoring the prognostic importance of these factors, particularly in Bismuth type IV tumors.

**Abstract:**

Introduction: Perihilar cholangiocarcinoma is a rare malignancy of the biliary tract, for which surgery remains the treatment of choice. However, even after radical resection, the prognosis is poor. In addition to tumor size, depth of invasion and nodal/metastatic status, the TNM classification includes additional parameters such as perineural (Pn), lymphangio (L) and vascular (V) invasion. The prognostic impact of these factors is not yet fully understood. The aim of this study was to investigate the influence of these parameters on overall survival after resection of perihilar cholangiocarcinoma. Material and Methods: Data from all patients who underwent surgical exploration for perihilar cholangiocarcinoma between January 2013 and December 2023 were included into an institutional database. The impact of perineural, lymphangio and vascular invasion on overall survival was analyzed. Results: Over the 11-year period, a total of 214 patients underwent surgical exploration for perihilar cholangiocarcinoma. Curative intended resection was possible in 168 patients (78.5%). Perineural invasion, lymphangio invasion and vascular invasion were present in 79.2%, in 17.3% and in 14.3% of patients, respectively. Cross tabulation revealed a significant association between the presence of L1 and V1 (*p* = 0.006). There was also a significant association of Pn1, L1, and V1 with R-status (*p* = 0.010; *p* = 0.006 and *p* ≤ 0.001). While V1 was associated with significantly worse overall survival across the entire cohort, Pn1 alone showed only a tendency towards worse overall survival without reaching statistical significance. In Bismuth type IV, both L1 and V1, but not Pn1, were significantly associated with worse overall survival (*p* = 0.001; *p* = 0.017 and *p* = 0.065). Conclusions: Perineural invasion is very common in perihilar cholangiocarcinoma. Although Pn1 was associated with a tendency toward worse survival, it did not reach statistical significance. In contrast, vascular invasion significantly worsened overall survival in the entire cohort, and lymphangio invasion was linked to worse overall survival in Bismuth type IV tumors. The combination of perineural invasion with positivity of more than one additional factor (either L or V) was also associated with worse overall survival. In patients with Bismuth type IV, these pathological markers appeared to have even greater prognostic relevance.

## 1. Introduction

Perihilar cholangiocarcinoma is defined as an adenocarcinoma of the biliary tract located between the second-degree bile ducts and the insertion of the cystic duct into the common bile duct. Surgical resection is the treatment of choice. The literature reports a median overall survival of 30 to 56 months in patients who have undergone potentially curative surgery [1,2]. 

However, the tumor’s location in close proximity to important structures in the liver hilum, such as the liver artery and portal vein, often makes curative resections technically challenging. To achieve an R0 situation, extended hepatic resections with vascular reconstructions often need to be performed [2]. However, even after potentially curative surgery, with or without vascular resection and reconstruction, the prognosis remains poor due to the infiltrative character of this tumor and its tendency for early hepatofugal and hepatopetal metastatic spread [2,3]. 

A wide range of factors has already been analyzed to predict outcomes following radical resection. In addition to tumor size, and nodal and metastatic status, the TNM classification includes other parameters such as perineural, lymphangio, and vascular invasion [4,5]. The prognostic relevance of these factors—particularly perineural invasion—has been investigated in several different tumors. However, there is limited data focusing specifically on perihilar cholangiocarcinoma, despite reports that this tumor shows a particular tendency to invade the perineural tissue surrounding vessels directly. 

Therefore, in this study, we examined the impact of perineural (Pn), lymphangio (L), and vascular (V) invasion on overall survival in a large, European single-center cohort of patients with resected perihilar cholangiocarcinoma.

## 2. Material and Methods

This study was designed as a retrospective single-center analysis. Data from all patients who underwent surgical exploration for perihilar cholangiocarcinoma between January 2013 and December 2023 were included in an institutional database. Patients under 18 years of age, those with tumors other than histologically proven perihilar cholangiocarcinoma, and those with benign conditions (e.g., IgG4-related diseases) were excluded from the analysis. All patients proved informed consent, allowing their data and follow-up information to be collected anonymously and used for scientific analyses. In accordance with the federal state laws (§36 and §37 of the state hospital laws) and the independent ethics committee of Rhineland-Palatinate, no ethical approval was required for this study.

For staging and surgical planning, either magnetic resonance tomography (MRI/MRCP) or computer tomography (CT) was preferred. Some patients presented with imaging already performed at referring centers or in outpatient settings. We accepted these external scans if they met the required quality standards. Biliary drainage was primarily performed using ERCP and stenting or alternatively by inserting a percutaneous transhepatic cholangiography drainage (PTCD). Prior to surgery, bilirubin levels were typically reduced to below 2 mg/dL, though in individual cases, levels up to 4 mg/dL were accepted. All cases were discussed in an interdisciplinary tumor conference, which included experienced hepatobiliary surgeons, radiologists, pathologists and oncologists. Surgical explorations and resections were performed by a dedicated team of hepatobiliary surgeons specializing in complex liver surgeries. Since 2008, the center has maintained a stable team of hepatobiliary surgeons who follow a standardized surgical approach. Following the publication of the BILCAP trial in 2017, patients have received adjuvant treatment with Capecitabine for six months, starting in 2018 [6].

Perineural, lymphangio, and vascular invasion were analyzed in histopathological specimens according to the eighth edition of the TNM classification [5]. Pathologists with a high level of expertise in hepato-pancreato-biliary cancers examined the resection specimens. Both ductal and radial margins were assessed to evaluate resection margins. Specimens were classified as R0 if tumor cells extended to just below the ink-marked resection margin (< 0.1 cm). Microscopically, several cell layers were required between the resection specimen and the ink marking. The Bismuth classification was determined based on pathological findings in the resection specimen. Postoperative follow-up was conducted every three months for at least two years after resection, with CT or MRI scans alternating with ultrasounds at least every six months.

Statistical analyses were performed using SPSS 23 (SPSS Inc., released 2014, IBM SPSS Statistics for Windows, Version 23.0, IBM Armonk, NY, USA: IBM Corp.). The chi^2^ test was used to analyze categorical data in cross-tabulations. Overall survival was assessed using the Kaplan–Meier method and log-rank test, while multivariate analysis was conducted with the Cox regression model. A *p*-value of <0.05 was considered statistically significant.

## 3. Results

During the 11-year survey period, 214 patients underwent surgery for perihilar cholangiocarcinoma and were included in this study. The median age was 70.0 years (range: 32.2–85.7 years), with a gender distribution of 82 female (38.3%) and 132 male (61.7%) patients. Five patients received neoadjuvant treatment: two with Folfirinox, one with Gemcitabin/Cisplatin, one with Gemcitabin/Durvalumab and one with Gemcitabin/Cisplatin/Durvalumab.

Out of 214 patients, 168 patients (78.5%) underwent tumor resection, while 46 patients had surgical exploration only. The resections performed included right or left hemihepatectomy (*n* = 93), extended right or left hemihepatectomy (*n* = 39), mesohepatectomy (*n* = 6), central bisegmentectomy (*n* = 4) or resection of extrahepatic bile ducts (*n* = 26).

The primary reasons for irresectability were peritoneal carcinomatosis (*n* = 21), major involvement of the hepatic artery or other structures of the liver hilum not amenable to resection and reconstruction (*n* = 16). As well as other reasons, such as insufficient quality or volume of the remaining liver parenchyma, even after adequate biliary decompression and intended volume modulation, or extended infiltration of adjacent structures, e.g., the duodenum, or a combination of these findings (*n* = 9). In two patients, final histology revealed an M1 situation. Both of these patients had an incidental tumor node (20 mm and 7 mm, respectively) identified in the right-sided hemihepatectomy specimen, distant from the bile duct resection margin. The T-, N- and R-status, as well as UICC classification for the 168 cases, are summarized in Table 1.

Based on the pathological findings in the resection specimen, the tumors were classified according to the Bismuth classification as follows: type I *n* = 11, type II *n* = 11, type IIIa *n* = 25, type IIIb *n* = 40, type IV *n* = 81.

A total of 133 out of 168 tumors (79.2%) showed perineural invasion. The results were similar within the Bismuth classification, ranging from 72.0% to 84.0% (Table 2). Lymphangio invasion was present in 17.3% of patients, and vascular invasion in 14.3%, as detailed in Table 2, according to the Bismuth classification.

Cross-tabulation of factors associated with perineural, lymphangio, and vascular invasion is presented in Table 3. Significant results were observed for the following combinations: L1/L0 vs. V1/V0 (*p* = 0.006) and Pn1/Pn0, L1/L0, V1/V0 vs. R-status (*p* = 0.010, *p* = 0.006 and *p* ≤ 0.001). No associations were found between P-, L- or V-status and grading or M-status. Nodal status was significantly associated with lymphangio invasion (*p* < 0.001).

The mean follow-up period for the entire cohort was 17.2 months (range: 1–99.5 months). The overall survival rates for all patients at 1, 3 and 5 years were 79%, 51% and 39%, respectively. Patients with vascular invasion had significantly worse overall survival compared to those without vascular invasion (V1 vs. V0: *p* = 0.032) (Figure 1E). No statistically significant difference in survival was observed for patients with or without perineural invasion (Pn1 vs. Pn0: *p* = 0.752) or lymphangio invasion (L1 vs. L0: *p* = 0.059) (Figure 1A,C). In the subgroup of patients with Bismuth type IV tumors, overall survival was significantly better for those without lymphangio or vascular invasion (L1 vs. L0: *p* = 0.001 and V1 vs. V0: *p* = 0.017) (Figure 1D,F).

We further conducted an analysis for R0-resections, which yielded results similar to those in the complete cohort. Again, V1 proved to be a significant prognostic factor. In patients with Bismuth type IV tumors, overall survival was significantly worse for those with lymphangio and/or vascular invasion (L1 vs. L0: *p* = 0.003 and V1 vs. V0: *p* = 0.016) (Figure 2D,F). However, survival for patients with perineural invasion and lymphangio invasion in the complete R0-cohort did not reach statistical significance (Pn1 vs. Pn0: *p* = 0.830 and L1 vs. L0: *p* = 0.143) (Figure 2A,C).

We conducted an additional analysis comparing overall survival in patients with R0-resection of perihilar cholangiocarcinoma, categorized as L0/V0/Pn0, versus those with one positive factor (either L1, V1, or Pn1) and those with two or more positive factors (Figure 3). The results indicated worse overall survival for patients with one positive factor (either L1, V1, or Pn1) and for those with ≥ two positive factors in both the complete cohort (Figure 3A) and in patients with a Bismuth type IV tumor (Figure 3B). However, these findings did not reach statistical significance (*p* = 0.082 and *p* = 0.055).

Several parameters were tested using univariate analysis. Some factors were found to be significantly associated with overall survival and were subsequently included in the multivariate Cox regression analysis (Table 4). N-status, grading and UICC status emerged as independent predictors of overall survival.

## 4. Discussion

Surgical resection with the goal of achieving an R0 situation is the gold standard for the treatment of resectable perihilar cholangiocarcinoma [7]. However, at the time of diagnosis, most patients are not eligible for surgery due to locally advanced or, less commonly, metastatic disease [8,9]. When surgery is feasible, the type of the resection is orientated by both the longitudinal extent of the tumor within the bile duct and the lateral spread, including possible invasion of the hepatic artery and portal vein. As a result, the surgical strategy is closely linked to the tumor’s cross-sectional appearance, as represented by the Bismuth classification, and the radial extent within the bile duct, reflected by the T-category. Needless to say, locoregional lymph node removal is standard practice. However, survival after potentially curative resection is influenced by more than just the tumor size, depth of invasion, and the nodal and metastatic status as defined by the TNM classification. Other factors, such as the tumor’s grading and perineural invasion, also play a key role in predicting outcomes [5].

In various studies, perineural invasion in malignant tumors has been shown to be associated with reduced overall survival rates [10,11,12]. In our study, there was a high prevalence of perineural invasion in perihilar cholangiocarcinoma (79.2%). Pn positivity was associated with a tendency to worse overall survival but did not reach statistical significance (*p* = 0.752). This finding is somewhat surprising, as the negative prognostic impact of Pn positivity has been demonstrated in other biliary tumors, such as intrahepatic cholangiocarcinoma [13]. The most likely explanation for our result is the high prevalence of Pn positivity in our cohort, coupled with the relatively small number of patients on the other hand. Thus, maybe the group of Pn-negative tumors was too small to achieve statistical significance. Another possible explanation is the increased number of patients receiving adjuvant therapy during the study period, potentially mitigating the negative prognostic effect of Pn positivity. Similarly, Kim et al. were unable to show a statistically significant association between perineural invasion and poorer survival in a group of 91 resected distal cholangiocarcinoma, provided that adequate lymph node dissection (>11) was performed [10]. In contrast, lymphangio invasion showed a clear tendency toward worse overall survival in the complete cohort, though it did not reach statistical significance (*p* = 0.059). However, in patients with perihilar cholangiocarcinoma Bismuth type IV, the association was statistically significant (*p* = 0.001). It is not surprising that L1 was significantly associated with a higher rate of nodal positivity. In our study, vascular invasion had a significant impact on overall survival in the entire cohort (*p* = 0.032), and particularly in patients with Bismuth type IV tumors (*p* = 0.017). This result was also observed when more than one positive factor (either L1 or Pn1) was present. Nevertheless, patients who underwent resection still had better survival compared to those with unresected tumors, who had a median survival of 10 months (range: 0–40 months) in our cohort. This observation is consistent with findings by Bae et al. [14].

Numerous studies have demonstrated the prognostic impact of achieving a negative resection margin in perihilar cholangiocarcinoma [2]. According to some studies, R0 resections are reported in 52–67% of cases [15,16]. In our study, we achieved a slightly higher R0 rate of 76.8%, yet approximately one-fourth of resections were still classified as R1. Careful pathological examination of the resection specimen and critical analysis of the histopathological results revealed that the highest risk for an R1 resection lies in the liver hilum (circumferential margin) and at the liver cutting surface [17]. Consequently, more aggressive surgical procedures—such as extended right or left hepatectomies or even trisectionectomies—have been attempted to reduce the risk of a positive parenchymal margin. Similarly, vascular resections and reconstructions are performed to address vascular invasion and minimize the risk for a positive circumferential margin [18,19,20,21]. While these aggressive approaches may increase surgical radicality, they also come at the cost of higher perioperative morbidity and mortality. This observation has sparked a debate about whether left or right hepatectomy is preferable in cases where the tumor extent permits both possibilities [20,22]. Both surgical attempts aim to enhance radicality in the hepatic hilum and liver, targeting lateral and hepatopetal tumor growths and spread, respectively. However, perihilar cholangioarcinoma also tends to spread hepatofugally. In addition to intraductal tumor growth, perineural, lymphangio, and vascular invasions are the primary sources of hepatofugal tumor spread. Unlike lymphangio invasion and lymphatic spread, which occurs discontinuously, perineural invasion allows a continuous path of tumor growth [23]. Of particular interest is tumor spread toward and behind the pancreas, where early local tumor recurrences frequently occur [24,25,26]. Considering the hepatofugal spread of the tumor, it becomes evident that there is a limit to the radicality achievable, even with extended surgery in the liver hilum. This has prompted consideration of less aggressive liver resections. Notably, Bröring et al.—in a small, highly selective case series—demonstrated that parenchyma-preserving hepatectomies were associated with lower postoperative morbidity and mortality while offering similar overall survival outcomes compared to extended liver resections [27]. Benzing et al. reached a similar conclusion for patients with nodal involvement, who might benefit from less extensive surgery [28]. Although these data are based on small sample sizes and do not allow any valid conclusion, it is possible that parenchyma-preserving resections could play a more important role in selected patients in the future, even in perihilar cholangiocarcinoma [29].

This is also due to recent progress in adjuvant treatment for perihilar cholangiocarcinoma, which may have biased outcomes. The BILCAP trial has led to changes in postoperative protocols, making adjuvant treatment with Capecitabin the current standard of care [30,31,32]. Zhao et al. demonstrated that adjuvant chemotherapy is an independent prognostic factor for overall survival in patients with perineural invasion in perihilar cholangiocarcinoma [33]. Similarly, Murakami et al. found that adjuvant chemotherapy improves overall survival in patients with perineural invasion, with a 5-year survival of 33% for those receiving adjuvant therapy compared to 21% for those without [11]. Despite these advancements, the need for more effective postoperative adjuvant treatments remains clear, given the persistently poor survival rates. The presence of lymphangio and vascular invasion, especially in Bismuth type IV tumors, highlights the need for adjustments and extensions to current adjuvant therapy protocols. Perineural-, lymphangio- and vascular invasion may also influence the choice of adjuvant therapy in the future. Further research in this area is warranted, as additional treatments, including immunotherapy, could potentially supplement adjuvant therapy in high-risk patients (Pn+/L+/V+). Oh et al. demonstrated improved overall survival after adjuvant therapy with gemcitabine plus durvalumab in patients with advanced biliary tract cancer [6]. In contrast, the role of preoperative therapy in perihilar cholangiocarcinoma remains in its early stages. While some studies suggest that preoperative chemotherapy may increase the chance of achieving R0 resection in patients with borderline or unresectable perihilar cholangiocarcinoma, its impact on overall survival still needs to be clarified [34,35].

It is also important to note that patient comorbidities cannot be ruled out as confounding factors, which may have contributed to the lack of statistical significance in some subgroups. These comorbidities were not thoroughly investigated across the various subgroups and could represent potential sources of bias. This influence should be further explored in future studies.

Given the retrospective nature of our study, there are some limitations. The relatively small number of patients in certain subgroups, particularly those with lymphangio and vascular invasion, may have prevented us from reaching conclusive and statistically significant results. Additional studies with larger sample sizes with a prospective design are necessary to better assess the prognostic impact of perineural, lymphangio, and vascular invasion.

## 5. Conclusions

Perineural invasion is highly prevalent in perihilar cholangiocarcinoma. Although Pn positivity was associated with a tendency toward worse survival, it did not reach statistical significance. In contrast, vascular was significantly associated with worse overall survival in the entire cohort, while lymphangio invasion had a significant impact in patients with Bismuth type IV tumors. The combination of perineural invasion and positivity of more than one additional factor (either L or V) also resulted in worse overall survival. In patients with Bismuth type IV tumors, these pathological markers appeared to have even greater prognostic relevance.

## Figures and Tables

**Figure 1 cancers-16-03463-f001:**
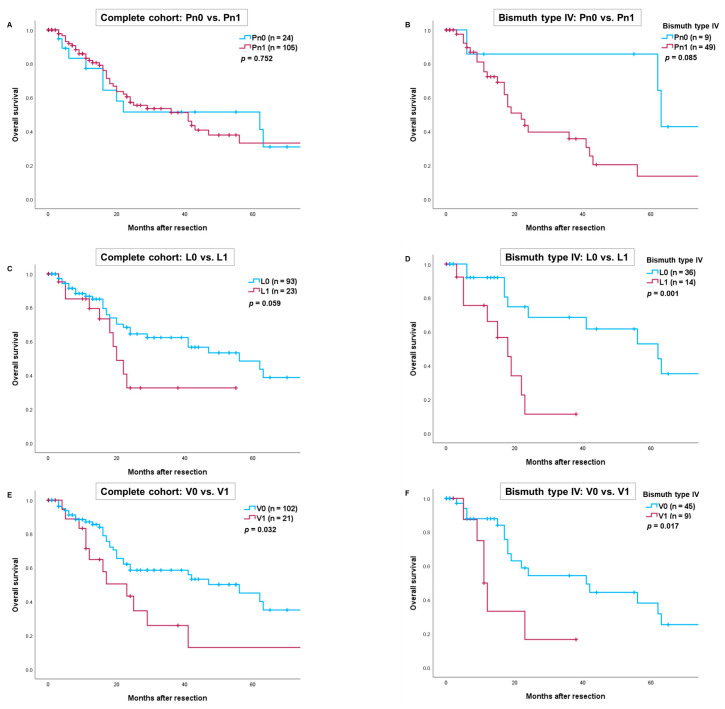
Comparison of overall survival after resection of perihilar cholangiocarcinoma depending on perineural invasion in the complete cohort (**A**) and patients with Bismuth type IV (**B**). Comparison of overall survival depending on lymphangio invasion in the complete cohort (**C**) and patients with Bismuth type IV (**D**). Comparison of overall survival depending on vascular invasion in the complete cohort (**E**) and patients with Bismuth type IV (**F**). Perioperative deaths were excluded from the statistical analysis.

**Figure 2 cancers-16-03463-f002:**
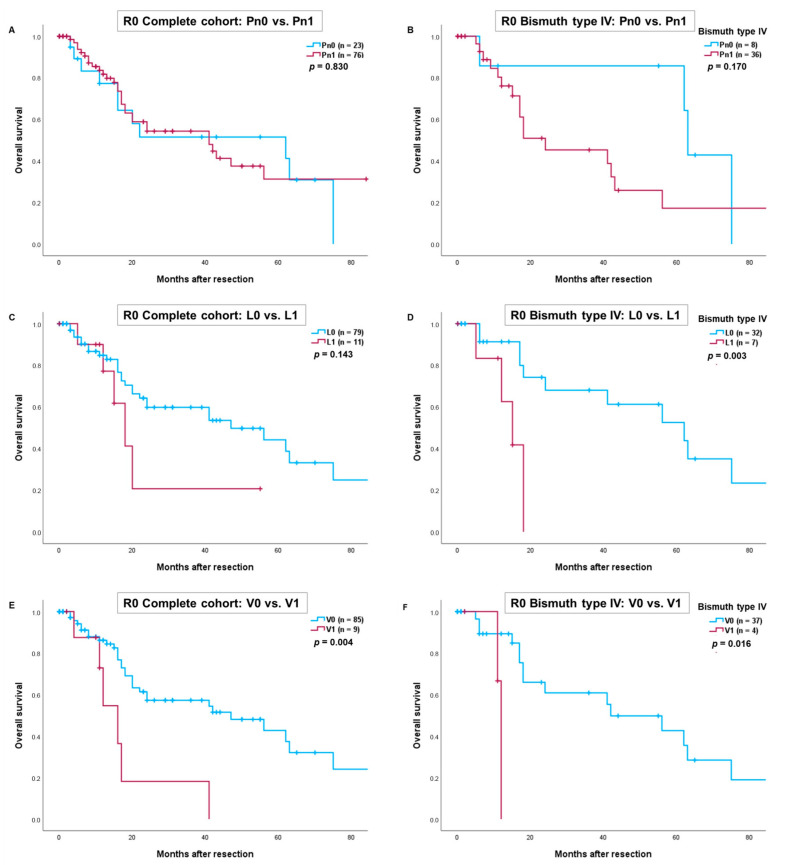
Comparison of overall survival after R0-resection of perihilar cholangiocarcinoma depending on perineural invasion in the complete cohort (**A**) and patients with Bismuth type IV (**B**). Comparison of overall survival depending on lymphangio invasion in the complete cohort (**C**) and patients with Bismuth type IV (**D**). Comparison of overall survival depending on vascular invasion in the complete cohort (**E**) and patients with Bismuth type IV (**F**). Perioperative deaths were excluded from the statistical analysis.

**Figure 3 cancers-16-03463-f003:**
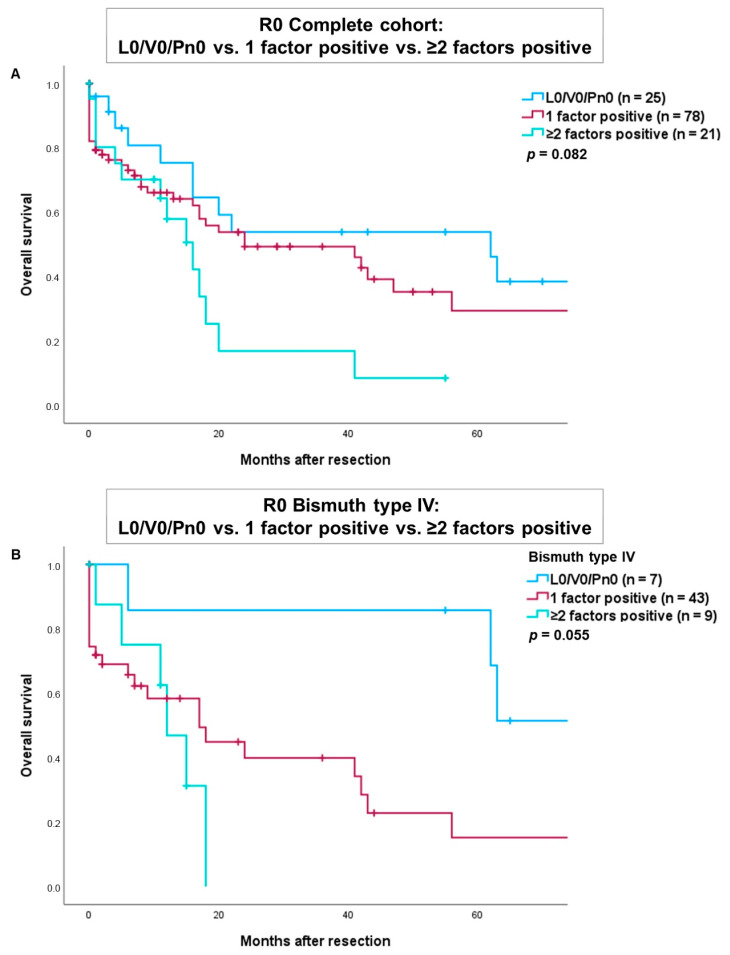
Comparison of overall survival after R0-resection of tumors with L0/V0/Pn0 versus one positive factor (either L1, V1 or Pn1) versus ≥ two positive factors in the complete cohort (**A**) and patients with a Bismuth type IV tumor (**B**).

**Table 1 cancers-16-03463-t001:** Overview of T-, N- and R-status, as well as UICC classification, for 168 resected patients. In patients classified as Nx, no lymph nodes were examined in the specimen, and therefore, no UICC classification could be assigned.

T-Status		N-Status		R-Status		UICC	
T1	10	N0	104	R0	129	I	6
T2a	51	N1	50	R1	38	II	89
T2b	91	N2	6	Rx	1	IIIA	8
T3	13	Nx	8			IIIB	4
T4	3					IIIC	48
						IVA	6
						Nx	8

**Table 2 cancers-16-03463-t002:** Overview of Pn, L and V-status in all patients, categorized by Bismuth type I to IV. (n.d. = no data available).

		Type I (*n* = 11)	Type II (*n* = 11)	Type IIIa (*n* = 25)	Type IIIb (*n* = 40)	Type IV (*n* = 81)
Pn0	*n* = 26	1	2	5	8	10
Pn1	*n* = 133	9	9	18	29	68
n.d.	*n* = 9	1	-	2	3	3
L0	*n* = 114	10	7	17	30	50
L1	*n* = 29	0	3	4	5	17
n.d.	*n* = 25	1	1	4	5	14
V0	*n* = 126	10	7	18	31	60
V1	*n* = 24	0	3	4	6	11
n.d.	*n* = 18	1	1	3	3	10

**Table 3 cancers-16-03463-t003:** Factors associated with perineural, lymphangio, and vascular invasion in cross-tabulation. Significant *p*-values are indicated in bold. ^a^ Eight patients with Nx were excluded from this analysis. ^b^ Six patients with Gx were excluded. ^c^ One patient with Rx was excluded.

	Pn1	Pn0	*p*	L1	L0	*p*	V1	V0	*p*
	*n* = 133	*n* = 26		*n* = 29	*n* = 114		*n* = 24	*n* = 126	
**T-status**			0.660			0.282			**<0.001**
T1 + T2 *n* = 153	119	24		28	103		14	121	
T3 + T4 *n* = 16	14	2		1	11		10	5	
**N-status ^a^**			0.230			**<0.001**			0.365
N0 *n* = 104	79	18		9	78		13	79	
N1 *n* = 56	48	6		19	29		10	40	
**M-status**			0.529			0.473			0.534
M0 *n* = 166	131	26		29	112		24	124	
M1 *n* = 2	2	-		-	2		-	2	
**Pn-status**			-			0.069			0.189
Pn0 *n* = 26	-	-		2	24		2	24	
Pn1 *n* = 133	-	-		27	87		22	99	
**L-status**			0.069			-			**0.006**
L0 *n* = 114	87	24		-	-		10	104	
L1 *n* = 29	27	2		-	-		8	21	
**V-status**			0.189			**0.006**			-
V0 *n* = 126	99	24		21	104		-	-	
V1 *n* = 24	22	2		8	10		-	-	
**Grading ^b^**			0.283			0.205			0.989
G1 + G2 *n* = 117	88	19		16	79		16	84	
G3 *n* = 45	41	5		11	31		7	37	
**R-status ^c^**			**0.010**			**0.006**			**<0.001**
R0 *n* = 129	96	25		16	93		9	104	
R1 *n* = 38	36	1		12	21		14	22	

**Table 4 cancers-16-03463-t004:** Univariate and multivariate analysis. Significant parameters are indicated in bold. Parameters with *p* < 0.05 were included in the multivariate analysis. Significant *p*-values are indicated in bold. OS = overall survival, HR = hazard ratio, 95% CI = 95% confidence interval. ^a^ Nine patients with Nx were excluded from this analysis. ^b^ One patient with Rx was excluded from this analysis. ^c^ Six patients with Gx were excluded from this analysis.

		Kaplan–Meier	Multivariate Cox Regression		
		OS	OS		
			HR	95% CI	*p*-Value
**T-status**	T1 + T2a + T2b/T3 + T4	0.132			
**N-status ^a^**	N0/N+	**0.008**	0.369	0.089–1.533	0.170
**V-status**	V0/V1	0.216			
**L-status**	L0/L1	0.131			
**Pn-status**	Pn0/Pn1	0.543			
**M-status**	M0/M1	0.540			
**R-status ^b^**	R0/R1	0.704			
**Grading ^c^**	G1 + G2/G3	**<0.001**	1.572	1.240–1.994	**<0.001**
**UICC stage**	I + II/IIIa + IIIb + IIIc + IVa + IVb	**0.006**	1.773	1.111–2.830	**0.016**

## Data Availability

The datasets used and analyzed during the current study are available from the corresponding author upon reasonable request.

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
