# Peer review of "Influence of Perineural (Pn), Lymphangio (L) and Vascular (V) Invasion on Survival after Resection of Perihilar Cholangiocarcinoma"

_cancers, 2024, doi:10.3390/cancers16203463_

Round 1
Reviewer 1 Report
Comments and Suggestions for Authors
The authors present a retrospective study analyzing surgical margins for patients with perihilar carcinoma. The methodology is clearly outlined and the data analysis with figures and tables is easy to interpret. While the focus of the study is based on perineural, lymphangio, and vascular invasion the authors don't address adjuvant or neoadjuvant therapies that the patients may have received. With a survival based study, this information is important to include.
Comments on the Quality of English LanguageThe english is average quality, there are a few areas where the word selection needs to be changed.
Author Response
Comments for the Reviewers and Editors
We want to thank the reviewers for their time and effort for reviewing our manuscript. Changes are marked green within the manuscript.
Reviewer 1
The authors present a retrospective study analyzing surgical margins for patients with perihilar carcinoma. The methodology is clearly outlined and the data analysis with figures and tables is easy to interpret. While the focus of the study is based on perineural, lymphangio, and vascular invasion the authors don't address adjuvant or neoadjuvant therapies that the patients may have received. With a survival based study, this information is important to include.
Comment:
A number of five patients received neoadjuvant treatment (two patients Folfirinox, one patient Gemcitabin/Cisplatin, one patient Gemcitabin/Durvalumab and one patient Gemcitabin/Cisplatin/Durvalumab). (row 126-128)
Following the publication of the BILCAP trial in 2017, patients received adjuvant treatment with Capecitabine for six months starting in 2018. (row 103-105)
Thank you again for your comments and help to improve our manuscript.
Rabea Margies, Fabian Bartsch and Hauke Lang

Reviewer 2 Report
Comments and Suggestions for Authors
- The objective of the study is clearly stated, but the rationale for focusing specifically on the combination of perineural, lymphangio, and vascular invasion in perihilar cholangiocarcinoma could be expanded. Why are these specific factors of particular interest compared to others?
- The study mentions that certain findings did not reach statistical significance. It would be helpful to discuss in more depth why these findings might still be clinically relevant despite the lack of statistical significance. This would add nuance to the interpretation of the data.
- The manuscript mentions that the sample size may have been too small to detect significant differences in some analyses. Conducting and reporting a power analysis could strengthen the study by demonstrating whether the sample size was adequate to detect meaningful differences.
- The discussion could be enriched by addressing potential confounding factors that might have influenced the outcomes, such as variations in surgical technique, patient comorbidities, or adjuvant therapies.
- While the subgroup analysis for Bismuth type IV tumors is interesting, it is not entirely clear why this particular subgroup was chosen. Providing a rationale for focusing on this subgroup would clarify its importance.
- The study spans an 11-year period, yet the potential changes in surgical techniques, patient management, or diagnostic criteria over time are not discussed. Addressing how these factors might have influenced the results would provide context for interpreting the findings.
- The tables and figures effectively summarize the data, but some of the figures, particularly those comparing survival curves, could benefit from clearer labels and more detailed legends to make them more self-explanatory.
- While the limitations of the study are mentioned, they could be more thoroughly explored. Specifically, the retrospective design of the study and its implications for causality should be discussed in greater detail.
- The discussion could be expanded to include more about how the findings might influence clinical decision-making. For instance, how might these findings change the management of patients with perihilar cholangiocarcinoma?
- The conclusion mentions the need for future studies, but it would be beneficial to provide more specific suggestions for what future research should focus on, such as prospective studies or trials that include adjuvant therapies.
- The authors need to add the serum analysis of normal control and other groups.
- HE and IHC staining of the samples are required.
- The authors need to CT scan data
- The author needs to add western blot results along with markers.
Minor editing of English language required.
Author Response
Comments for the Reviewers and Editors
We want to thank the reviewers for their time and effort for reviewing our manuscript. Changes are marked green within the manuscript.
Reviewer 2
- The objective of the study is clearly stated, but the rationale for focusing specifically on the combination of perineural, lymphangio, and vascular invasion in perihilar cholangiocarcinoma could be expanded. Why are these specific factors of particular interest compared to others?
Comment:
Various factors regarding overall survival have already been analysed in various tumor entities. Pn, L and V are established factors in daily clinical routine due to TNM-classification. To the best of our knowledge there are only rare data regarding the influence of these specific factors in perihilar cholangiocarcinoma as this tumor has the tendency to directly invade the perineural tissue of the vessels due to its location in the liver hilum. (Introduction row 72-79 and discussion row 217-228)
- The study mentions that certain findings did not reach statistical significance. It would be helpful to discuss in more depth why these findings might still be clinically relevant despite the lack of statistical significance. This would add nuance to the interpretation of the data.
Comment:
We kindly refer to discussion row 224 to 247. In these lines we go into this in detail and discuss the significance for the positivity of L and V. We also explain and discuss why, in our view, unexpectedly no significance could be achieved with positive perineural infiltration.
- The manuscript mentions that the sample size may have been too small to detect significant differences in some analyses. Conducting and reporting a power analysis could strengthen the study by demonstrating whether the sample size was adequate to detect meaningful differences.
Comment:
We refer to two previous studies with regard to this comment. Power analysis is mainly used in prospective studies, but not in retrospective designs. It is particularly popular to blame low power for non-significance. This point can be discussed, because power increases with decreasing significance of the test at constant effect size. Zhang et al. showed that such power analyses did not indicate true power for detecting statistical significance. (DOI: 10.1136/gpsych-209-100069). Dziak et al. came to the point that power analysis cannot provid a valid way of deciding whether a larger sample really is practically warranted, or whether the effect size is too small to be worth detecting. (DOI: 10.1007/s12144-018-0018-1)
From our point of view, in regard to our retrospective study design a power analysis will raise more questions than answering them.
- The discussion could be enriched by addressing potential confounding factors that might have influenced the outcomes, such as variations in surgical technique, patient comorbidities, or adjuvant therapies.
Comment:
Regarding adjuvant therapy there was a change in protocols after BILCAP trial in 2017. We added this as a potential influence/bias in our discussion. (row 287)
This is also because there has been some progress in adjuvant treatment of perihilar cholangiocarcinoma, which might have biased the outcomes. (row 287)
Regarding surgical techniques there is no relevant bias expected. We added this in our material and methods:
The center offers a stable team of hepatobiliary surgeons since 2008, who follow a uniform surgical concept. (row 102-103)
It should also be mentioned that patient comorbidities cannot be ruled out as confounding factors that some subgroups did not reach statistical significance. These were not conclusively investigated in the various subgroups and may represent possible bias. This influence should be investigated in future studies (row 307-310)
- While the subgroup analysis for Bismuth type IV tumors is interesting, it is not entirely clear why this particular subgroup was chosen. Providing a rationale for focusing on this subgroup would clarify its importance.
Comment:
In Germany/Europe, many people regard Bismuth type IV as irresectable or wrongly equate it with patients classified as T4 according to TNM-classification. For this reason, on the one hand, we wanted to show that many patients with Bismuth type IV tumors were resected. On the other hand, we wanted to show that the positivity of Pn, L and V represents an increased risk or a worse outcome due to the local spread of the tumor in the liver hilus in this subgroup of patients.
- The study spans an 11-year period, yet the potential changes in surgical techniques, patient management, or diagnostic criteria over time are not discussed. Addressing how these factors might have influenced the results would provide context for interpreting the findings.
Comment:
We kindly refer to comment number 4 and that there is a stable pre-, peri- and postoperative work flow in our single center over time, primarily due to the same head of department over the 11-year period of this study. A comment to changes due to the influence of adjuvant treatment is mentioned in our discussion; please also see the first part of comment 4.
- The tables and figures effectively summarize the data, but some of the figures, particularly those comparing survival curves, could benefit from clearer labels and more detailed legends to make them more self-explanatory.
Comment:
We have adapted the graphics of the survival curves with clearer labeling.
- While the limitations of the study are mentioned, they could be more thoroughly explored. Specifically, the retrospective design of the study and its implications for causality should be discussed in greater detail.
Comment:
We kindly refer to our discussion line 306-311 as well as the first part of comment 10. We also added a part regarding comorbidities, of course there could be some limitations, but unfortunately we did not focus on this part in the study. Nevertheless, in our opinion, the study stands out as a single center due to the high number of cases.
It should also be mentioned that patient comorbidities cannot be ruled out as confounding factors that some subgroups did not reach statistical significance. These were not conclusively investigated in the various subgroups and may represent possible bias. This influence should be investigated in future studies. (row 307-310)
- The discussion could be expanded to include more about how the findings might influence clinical decision-making. For instance, how might these findings change the management of patients with perihilar cholangiocarcinoma?
Comment:
We fully understand this point. According to our expectations, we would have expected a change in overall survival in patients with perineural infiltration. In the future, adjuvant therapy other than just Capecitabine mono may be necessary. The positivity of Pn, L and V may help to address this. In Bismuth type IV, this can indeed be discussed if L and V are positive.
The positivity of L and V, especially in Bismuth type IV tumors, clearly indicates the need for extensions and adjustments to adjuvant therapy. (row 295-297)
- The conclusion mentions the need for future studies, but it would be beneficial to provide more specific suggestions for what future research should focus on, such as prospective studies or trials that include adjuvant therapies.
Comment:
In order to increase the number of cases, a multicenter study makes sense. The major disadvantage, however, is the increase in inhomogeneity, which will result in a greater bias. Subgroups could, however, be filled with significantly larger numbers of cases. It will be extremely difficult to conduct randomized controlled trials in this setting, as recruitment will be difficult.
Perineural-, lymphangio- and vascular invasion could also have an influence on the choice of adjuvant therapy in the future. Further research should be sought in this area. If necessary, additional immunotherapy can supplement adjuvant therapy in risk constellations (Pn+/L+/V+). Oh et al. were able to demonstrate an improvement in overall survival after adjuvant therapy with gemcitabine plus durvalumab for patients with advanced biliary tract cancer [35]. (row 297-302)
- The authors need to add the serum analysis of normal control and other groups.
Comment:
Bilirubin levels prior to surgery should undergo a level of 2mg/dl. In individual cases a level up to 4mg/dl was also accepted. (row 97-99)
- HE and IHC staining of the samples are required.
Comment:
We kindly ask for further explanation of this comment and what added value can be expected after adding it to the publication with regard to the aim of this study.
- The authors need to CT scan data
Kindly refer to comment number 12
- The author needs to add western blot results along with markers.
Kindly refer to comment number 12
Thank you again for your comments and help to improve our manuscript.
Rabea Margies, Fabian Bartsch and Hauke Lang

Reviewer 3 Report
Comments and Suggestions for Authors
The authors report on their experience in a main German University Center with considerable expertise in the treatment of perihilar cholangiocarcinoma. They report after a detailed review on the outcomes of surgery (probably some 15-20 % of all comers with this pathology?) in a program with some 20 resectional surgeries annually. They correlated outcomes with perineural invasion, lymphangio and vascular invasion. Vascular invasion was prognostically significantly associated with bad outcome, lymph and vasc in combination also, perineural invasion showed a trend towards unfavorable outcone only, possibly due too the too limited number of patients to reach significance. .
They immediately confirm one issue, that even major centers have a big problem to recruit considerable numbers of patients for meaningful (retrospective) comparative outcome results (also addressed by professor Clavien's + many others 2021 publications reviewing cases in 24 referral centers).
This reviewer tries to identify what meaningful new information has come out of all this hard work. I am mostly missing any visionary perspective of what the implications of this study are. What can we do with this information, what can we detect or guess preoperatively and would it make us do different in our recommendations or new research avenues. The reader likes to know.
As Klatskin and others already noted, patients tend to die often due to the obstructive complications of the biliary system before malignancy takes over otherwise, and surgical resection tends above all to be associated with a bigger relatively symptom free survival in that respect and a period free of repeated biliary stenting etc.
I am somewhat uncomfortable that we do not learn anything other than a partly satisfactory analysis (the authors concur) of prognosticators of unfavorable pathology: What was the cause of death, how many patient had prior to and had afterwards biliary complications/cholangitis ? In making recommendations what is the picture that we are now giving to the reader?
Would the authors comment on the issues R0/R1 ( DeBellis et al 2022) How did they define their R, a hot issue in the literature of the last decade
Of interest: were they able to confirm in all patients malignancy prior to surgery? No imitators of malignancy (relevant not for outcome of malignancy but all over outcome of this type of resections and informed consent questions to patient)
Minor
in the proof forwarded the reference list has doubled references
Author Response
Comments for the Reviewers and Editors
We want to thank the reviewers for their time and effort for reviewing our manuscript. Changes are marked green within the manuscript.
Reviewer 3
The authors report on their experience in a main German University Center with considerable expertise in the treatment of perihilar cholangiocarcinoma. They report after a detailed review on the outcomes of surgery (probably some 15-20 % of all comers with this pathology?) in a program with some 20 resectional surgeries annually. They correlated outcomes with perineural invasion, lymphangio and vascular invasion. Vascular invasion was prognostically significantly associated with bad outcome, lymph and vasc in combination also, perineural invasion showed a trend towards unfavorable outcone only, possibly due too the too limited number of patients to reach significance. .
They immediately confirm one issue, that even major centers have a big problem to recruit considerable numbers of patients for meaningful (retrospective) comparative outcome results (also addressed by professor Clavien's + many others 2021 publications reviewing cases in 24 referral centers).
This reviewer tries to identify what meaningful new information has come out of all this hard work. I am mostly missing any visionary perspective of what the implications of this study are. What can we do with this information, what can we detect or guess preoperatively and would it make us do different in our recommendations or new research avenues. The reader likes to know.
Comment:
We can well understand the objection. Unfortunately, scientific reality does not always mean that the clinical workflow changes with the scientific results. We had also hoped that the results would be different, especially with positive perineural infiltration. The influence on the choice of adjuvant therapy is certainly of interest. Further studies may result in an adjustment (especially in the choice of immunotherapies) depending on the Pn/L/V status. However, further studies are definitely required for this.
As Klatskin and others already noted, patients tend to die often due to the obstructive complications of the biliary system before malignancy takes over otherwise, and surgical resection tends above all to be associated with a bigger relatively symptom free survival in that respect and a period free of repeated biliary stenting etc.
I am somewhat uncomfortable that we do not learn anything other than a partly satisfactory analysis (the authors concur) of prognosticators of unfavorable pathology: What was the cause of death, how many patient had prior to and had afterwards biliary complications/cholangitis?
Comment:
The causes of death were in descending order:
- Multiorgan failure, (2) liver failure, (3) sepsis, (4) bleeding, (5) heart failure
Unfortunately, biliary complications and cholangitis were not in focus of this analysis, so that we are unable to make a detailed statement on this.
In making recommendations what is the picture that we are now giving to the reader?
Comment:
Future studies need to be conducted to improve adjuvant therapies adapted to perineural, lymphangio and vascular status. We have also defined risk constellations, especially in patients with Bismuth type IV.
Perineural-, lymphangio- and vascular invasion could also have an influence on the choice of adjuvant therapy in the future. Further research should be sought in this area. If necessary, additional immunotherapy can supplement adjuvant therapy in risk constellations (Pn+/L+/V+). Oh et al. were able to demonstrate an improvement in overall survival after adjuvant therapy with gemcitabine plus durvalumab for patients with advanced biliary tract cancer [35]. (row 297-302)
Would the authors comment on the issues R0/R1 (DeBellis et al 2022) How did they define their R, a hot issue in the literature of the last decade
Comment:
A very important point of view. Thank you for mentioning it. Fortunately, our center benefits from pathologists with high expertise in hepato-pancreato-biliary specimen. Regarding resection margins, both ductal and radial margin were taken into account. The specimen was classified as R0 if the tumor formations extended to just below the ink-marked resection margin (<0,1cm). Microscopically, it was necessary that several cell layers were present between resection specimen and ink marking. We added a short section on resection margins within material and methods as well as in discussion. (Material and Methods row 107-112 and Discussion row 254-259)
Of interest: were they able to confirm in all patients malignancy prior to surgery? No imitators of malignancy (relevant not for outcome of malignancy but all over outcome of this type of resections and informed consent questions to patient)
Comment:
Patients with tumors other than histologically proven perihilar cholangiocarcinoma, as well as benign entities (e.g. IgG4), were excluded from analysis. (row 87-88)
Nevertheless, only 10-20% were histologically proven prior to surgery. Most patients underwent surgical exploration due to the suspicion of perihilar cholangiocarcinoma.
Minor
in the proof forwarded the reference list has doubled references
Comment:
Reference list has been corrected.
Thank you again for your comments and help to improve our manuscript.
Rabea Margies, Fabian Bartsch and Hauke Lang

Round 2
Reviewer 1 Report
Comments and Suggestions for Authors
The authors have adequately addressed my comments.
Comments on the Quality of English LanguageEnglish quality is good, but it would benefit from a review.
Reviewer 2 Report
Comments and Suggestions for Authors
The authors are strongly recommended to add HE, IHC staining and Western blots images. Also, Add CT scan. The authors need to quantify the HE and IHC staining. Also, quantify the western blots too.
Comments on the Quality of English LanguageMinor editing of English language required.
Reviewer 3 Report
Comments and Suggestions for Authors
I have no new comments
